# Evo-NAS: Evolutionary-Neural Hybrid Agent for Architecture Search

## Abstract

Neural Architecture Search has shown potential to automate the design of neural networks. Deep Reinforcement Learning based agents can learn complex architectural patterns, as well as explore a vast and compositional search space. On the other hand, evolutionary algorithms offer higher sample efficiency, which is critical for such a resource intensive application. In order to capture the best of both worlds, we propose a class of Evolutionary-Neural hybrid agents (Evo-NAS). We show that the Evo-NAS agent outperforms both neural and evolutionary agents when applied to architecture search for a suite of text and image classification benchmarks. On a high-complexity architecture search space for image classification, the Evo-NAS agent surpasses the accuracy achieved by commonly used agents with only 1/3 of the search cost.

## 1 Introduction

Neural Networks (NN) have yielded success in many supervised and unsupervised learning problems. However, the design of state-of-the-art deep learning algorithms requires many decisions, normally involving human time and expertise. As an alternative, Auto-ML approaches aim to automate manual design with meta-learning agents. Many different approaches have been proposed for architecture optimization, including random search, evolutionary algorithms, Bayesian optimization and an approach based on a NN trained with Reinforcement Learning (RL).

RL-based neural architecture search yielded success in automatic design of state-of-the-art RNN cells (Zoph and Le, 2017), convolutional blocks (Zoph et al., 2017), activation functions (Prajit Ramachandran, 2018), optimizers (Bello et al., 2017; Wichrowska et al., 2017) and data augmentation strategies (Cubuk et al., 2018). Under this paradigm, the NN architecture is sampled from the agent's policy, which in turn is optimized to maximize the performance of the generated models on the downstream task.

Recent work has shown that evolutionary agents for architecture search can match or outperform deep RL methods (Real et al., 2018; So et al., 2019). Evolutionary agents can efficiently leverage a single good model by generating similar models via a mutation process. Deep RL methods generate new models by sampling from a learned distribution, and cannot easily latch on to patterns of a single model, unless it has been promoted multiple times through the learning process. However, evolution has the disadvantage of relying on heuristics or random sampling when choosing mutations. Unlike approaches based on a Neural Network, Evolutionary approaches are unable to learn patterns to drive the search.

The main contribution of this paper is to introduce a class of Evolutionary-Neural hybrid agents (Evo-NAS). We propose an evolutionary agent whose mutations are guided by a NN trained with RL. This combines the sample efficiency of evolutionary agents with the ability to learn complex patterns of NNs.

In Section 3, we give a brief description of state-of-the-art Neural and Evolutionary agents, and introduce the Evo-NAS agent in Section 4. In Section 5, we discuss the different training algorithms used for training the NN-based agents. Then, in Section 6, we present and discuss the properties of the proposed Evolutionary-Neural agent by applying it to a synthetic task. Finally, we apply Evo-NAS to architecture search benchmarks, showing that it outperforms both RL-based and Evolution-based algorithms on architecture search for a variety of text and image classification datasets.

## 2 RELATED WORK

In recent years, progress has been made in automating the design process required to produce state-of-the-art neural networks. Recent methods have shown that learning-based approaches can achieve state-of-the-art results on ImageNet (Zoph and Le, 2017; Liu et al., 2017a). These results have been subsequently scaled by transferring architectural building blocks between datasets (Zoph et al., 2017).

Some works explicitly address resource efficiency (Zhong et al., 2018; Pham et al., 2018; Liu et al., 2018; Xie et al., 2018), which is crucial, as architecture search is known to require a large amount of resources (Zoph and Le, 2017). In particular, ENAS (Pham et al., 2018) and DARTS (Liu et al., 2018) are computationally efficient techniques that have been employed to learn what segments of an architecture can be pruned. These methods can be applied on a narrow subset of the architecture search applications that satisfy the scalability and expressiveness constraints (Sciuto et al., 2019; Li and Talwalkar, 2019). Furthermore, gradient-based approaches like DARTS can be applied to search over parameters for which gradients from the task loss can be computed. These approaches cannot be applied to a broader set of architecture search applications such as those presented in this work.

Prior works have used evolutionary methods to evolve neural networks (Floreano et al., 2008; Stanley et al., 2009; Real et al., 2017; Conti et al., 2017; Xie and Yuille, 2017). Real et al. (2018) have shown that evolutionary agents applied to architecture search can match or outperform the standard deep-RL based agents. Such et al. (2017) have applied genetic algorithms to evolve the weights of the model.

RENAS (Chen et al., 2018) is an alternative RL-evolutionary hybrid that has been developed in parallel to Evo-NAS. RENAS also provides evidence that learning mutation patterns increases the efficiency of evolution. A key difference is that Evo-NAS learns a policy directly on the entire architecture search-space while RENAS learns a policy over the space of possible mutations applicable to a given parent model. Thus, the application of RENAS requires to define an additional mutation grammar and a more complex agent, e.g. composed of 4 distinct mutation generation sub-steps. The parent-to-child parameter sharing approach is orthogonal to the definition of the hybrid controller, and can be applied to Evo-NAS in future work.

Other than deep RL and evolution, different approaches have been applied to architecture search and hyperparameter optimization: bayesian optimization (Falkner et al., 2018), cascade-correlation (Fahlman and Lebiere, 1990), boosting (Cortes et al., 2016), deep-learning based tree search (Negrinho and Gordon, 2017), hill-climbing (Elsken et al., 2017), model-based approaches (Hutter et al., 2011), and random search (Bergstra and Bengio, 2012).

## 3 BASELINES

We compare the proposed Evo-NAS agent with alternative commonly used agents:

**Random Search (RS)** generates a new model by sampling every architectural choice from a uniform distribution over the available values. The performance of the RS agent gives a sense of the complexity of the task, and allows to estimate the quality gains to attribute to the use of a more complex agent.

**Neural Architecture Search (NAS)** (Zoph and Le, 2017) uses an RNN to perform a sequence of architectural choices that define a generated model. The resulting model is then trained on a downstream task, and its quality on the validation set serves as the reward for training the agent using policy gradient. In the following sections, we will refer to the standard NAS agent as the Neural agent.

**Aging Evolution Architecture Search** (Real et al., 2018) is a variant of the tournament selection method (Goldberg and Deb, 1991). A population of $P$ generated models is improved in iterations. At each iteration, a sample of $S$ models is selected at random. The best model of the sample, parent model, is randomly mutated to produce a new model, which is trained and added to the population. Each time a new model is added to the population, the oldest model in the population is discarded. In the following sections, we will refer to the Aging Evolution agent as the Evolutionary agent.

## 4 EVOLUTIONARY-NEURAL ARCHITECTURE SEARCH

The proposed Evo-NAS agent generates new models by mutating a parent model. At each iteration, the parent model is chosen by picking the model with highest reward among a random sample of $S$

models drawn from the population of the most recent $P$ models, similarly to the Evolutionary agent. However, unlike the Evolutionary agent, the mutations are not sampled at random among the possible architectural choices, but are sampled from distributions inferred by a recurrent neural network (RNN).

When mutating a parent model, for each architectural choice in the sequence defining the model, the Evo-NAS agent can either reuse the parent's value or sample a new one from the corresponding learned distribution. The decision whether to mutate a value is performed for each architectural choice independently. The probability of mutating a parent value is a hyperparameter of the agent, which we refer to as the mutation probability: $p$. The values for the architectural choices that need to be mutated are re-sampled from the distributions inferred by the underlying RNN. The RNN is designed to condition each distribution on all the prior architectural choices values, independently of whether the values have been ported from the parent's sequence or re-sampled. Formally, consider the parent sequence $(a_1, ..., a_n)$, and denote its prefix with $A_i = (a_1, ..., a_i)$ for $i \in \{1, ..., n\}$. Fix a possible child sequence $(a'_1, ..., a'_n)$, and similarly denote $A'_i = (a'_1, ..., a'_i)$. Denoting the parameters of the RNN by $\theta$, the probability of sampling the child sequence $A'_n$ under the policy $\pi_\theta$ defined by the RNN is

$$p(A'_n|A_n, \theta) = \prod_{i=1}^{n} p(a'_i|A_{i-1}, A'_{i-1}, \theta).$$

Let $M_1, ..., M_n$ be Bernoulli$(p)$-distributed random variables, where $M_i$ defines whether a mutation occurs at position $i$. Then, under the law of total probability, we can formalize the probability of generating a sequence $A'$ as:

$$p(A'_n|A_n, \theta) = \prod_{i=1}^{n} \sum_{m_i \in \{0,1\}} p(a'_i, m_i|A_{i-1}, A'_{i-1}, \theta) = \prod_{i=1}^{n} \left( (1-p) \cdot \mathbb{1}[a'_i = a_i] + p \cdot \pi_\theta(a'_i|A'_{i-1}) \right).$$

Evo-NAS sampling algorithm is represented in Figure 1. Refer to Appendix A for the pseudocode of the algorithm, and Appendix B for a comparison with the baseline approaches.

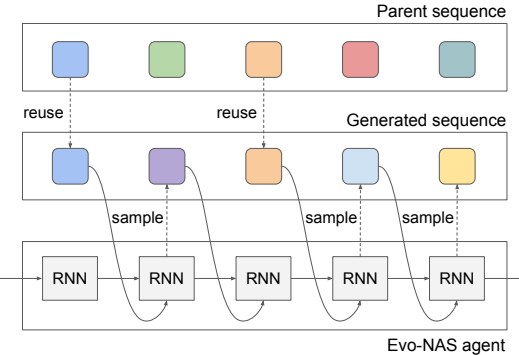

Figure 1: Evo-NAS agent sampling a sequence of architectural choices. Each of the colored blocks represents an architectural choice being set to a specific value. Each value of the generated sequence is sampled from the policy learned by the agent's neural network with probability $p$, or reused from the parent sequence with probability $1 - p$.

The Evo-NAS agent is initialized so that the distributions over the architectural choices are uniform. Thus, an initialized Evo-NAS agent produces random mutations as an Evolutionary agent. During training, the Evo-NAS's NN is updated by using policy gradient to maximize the expected quality metric achieved by the generated models on the downstream task. Therefore, the distributions over architectural choices will become more skewed with time to promote the architectural patterns of the good models. In contrast, an Evolutionary agent is unable to learn mutation patterns, since the distributions from which the mutations are sampled are constantly uniform.

We conjecture that Evo-NAS can retain the best qualities of both baselines: the sample-efficiency of Evolutionary agents, and the ability to learn complex patterns of Neural agents. We base this on the following observations. First, we note that the Evolutionary agent essentially biases the search toward the area around the best samples found so far; therefore, it acts with a prior belief that high

quality samples are often close to each other. As shown by (Real et al., 2018), this approach gives very strong performance on architecture search, suggesting that the clustering assumption is true for commonly used architecture search spaces. On the other hand, in high complexity search spaces, good models are likely to span only a low dimensional manifold. Under the assumption that the NN can learn the structure of this manifold, Evo-NAS can learn to restrict its local search to the manifold learned by the underlying NN. In Section 6 we empirically validate our conjecture, and show that Evo-NAS indeed outperforms both baseline algorithms.

## 5 TRAINING ALGORITHMS

We consider two alternative training algorithms for NN based agents such as Neural and Evo-NAS:

**Reinforce** (Williams, 1992). Reinforce is a standard policy-gradient training algorithm. It is often considered the default choice for applications where an agent needs to be trained to explore a complex search space to find the optimal solution, as is in the case of architecture search. This approach has the disadvantage of not being sample efficient, as it is not able to reuse samples. Reinforce is the training algorithm chosen in the original NAS paper (Zoph and Le, 2017).

**Priority Queue Training** (PQT) (Abolafia et al., 2018). With PQT, the NN gradients are generated to directly maximize the log likelihood of the best samples produced so far. This training algorithm has higher sample efficiency than Reinforce, as good models generate multiple updates. PQT has the simplicity of supervised learning, since the best models are directly promoted as if they constitute the supervised training set, with no need of reward scaling as in Reinforce, or sample probability scaling as in off-policy training.

## 6 EXPERIMENTS

### 6.1 SYNTHETIC TASK: LEARN TO COUNT

We compare the agents on a synthetic toy task designed to be complex enough such that random search would not succeed. Similarly to common architecture search spaces, it requires learning a sequential pattern. This task can be described as learning to count. The agent can sample sequence $\mathbf{a} = \langle a_1, a_2, \cdots, a_n \rangle$ of $n$ integer numbers, where each number is selected from the set $[1, n] \cap \mathbb{Z}$. The reward of a sequence $\mathbf{a}$ is defined as:

$$r(\mathbf{a}) = \frac{n+1}{a_1^2 + \sum_{k=1}^{n-1}(a_{k+1} - a_k)^2 + (a_n - (n+1))^2}$$

This reward is designed to encourage every two adjacent numbers to be close to each other, but also, to keep the first number small, and the last number large. The maximum reward of 1 is achieved by $\mathbf{a}^* = \langle 1, 2, \cdots, n \rangle$.

We compare Random, Evolutionary, Neural and Evo-NAS agents; additionally, for Neural and Evo-NAS agents we consider two alternative training algorithms: Reinforce and PQT.

We started by tuning the hyperparameters of all the agents independently, which we describe in Appendix C. The results of the comparison are shown in Figure 2.

We have found that PQT outperforms Reinforce for both the Neural and the Evo-NAS agent. For the Evo-NAS agent, the gain is especially pronounced at the beginning of the experiment. Thus, we conclude that training with PQT is more sample efficient than with Reinforce.

Now we turn to a comparison between: Random, Evolutionary, Neural (PQT) and Evo-NAS (PQT). The Evolutionary agent finds better models than the Neural agent during the initial 1000 samples, while in the second half of the experiment, the Neural agent outperforms the Evolutionary agent. Our interpretation is that Evolutionary agent's efficient exploitation allows to have a better start by mutating good trials. While the Neural agent needs 1000+ samples to learn the required patterns, after this is achieved it generates better samples than those generated by Evolutionary agent's random mutations. The results show that Evo-NAS agent achieves both the sample efficiency of Evolutionary approaches and the learning capability of Neural approaches. Evo-NAS initial fast improvement shows the ability to take advantage of the sample efficiency of evolution. Learning the proper

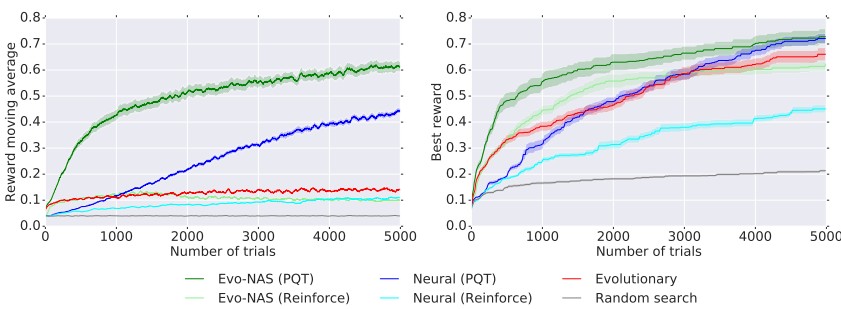

Figure 2: Reward moving average (left) and best reward (right) on "Learn to count" task averaged over 20 replicas. Shaded area represents 70% confidence interval.

mutation patterns allows it to keep outperforming the Evolutionary agent. The poor performance of the Random Search agent shows that the "Learn to count" task is non-trivial. We also see that the Neural agent would not have been able to catch up with the Evolutionary agent within the 5000 trials of this experiment if it was trained with Reinforce instead of PQT. For more details about the properties of this task see Appendix C.

## 6.2 NASBench

We perform the same comparison using NASBench (Ying et al., 2019), where the agents search for network architectures for CIFAR-10 classification task in a confined search space, which consists of approximately 423k unique architectures. We compare the same agents as in Section 6.1: Random, Evolutionary, Neural and Evo-NAS, and for Neural and Evo-NAS agents we consider training them with either Reinforce or PQT. We again tuned the hyperparameters for all the agents, and report the final values in Appendix D. We show the results of the comparison in Figure 3.

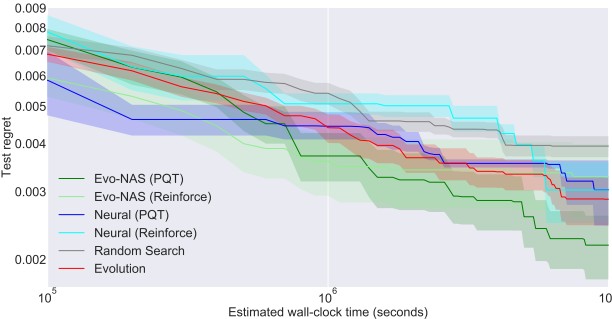

Figure 3: Reward moving average (left) and best reward (right) on NASBench task averaged over 64 replicas. Shaded area represents 70% confidence interval.

There are many similarities between Figures 2 and 3, but a key feature that sets them apart is that the quality achieved by Random Search is significantly better for NASBench than for "Learn to count". This is because a large percentage of architectures give relatively low test regret (less than 0.01).

Despite the difference mentioned above, we can draw a consistent conclusion from both lines of experiments. Evo-NAS and Evolutionary agents perform well at the beginning. Neural agent (trained with PQT) outperforms the Evolutionary agent when it collects a sufficient amount of samples to learn the pattern. On the other hand, Evo-NAS combines the qualities of the Neural and Evolutionary agents to deliver the best result.

## 6.3 TEXT CLASSIFICATION TASKS

We now compare the different agents on a real architecture search task. The Neural, Evolutionary and Evo-NAS agents are applied to the task of finding architectures for 7 text classification datasets.

Similarly to (Wong et al., 2018), we designed a medium complexity search space of common architectural and hyperparameter choices that define two-tower "wide and deep" models (Cheng et al., 2016). This search space is not designed to discover original architectures that set a new state-of-the-art for this type of tasks, but it allows to analyze the properties of the agents.

All the experiments in this section are run with a fixed budget: 30 trials are trained in parallel for 2 hours with 2 CPUs each. Choosing a small budget allows to run a higher number of replicas for each experiment to increase the significance of the results. It also makes the budget accessible to most of the scientific community, thus increasing the reproducibility of the results.

During the experiments, the models sampled by the agent are trained on the training set of the current text classification task, and the area under the ROC curve (ROC-AUC) computed on the validation set is used as the reward for the agent. To compare the generated models that achieved the best reward, we compute the ROC-AUC on the holdout testset as the final unbiased evaluation metric. For the datasets that do not come with a pre-defined train/validation/test split, we split randomly 80%/10%/10% respectively for training, validation and test set.

We validate the results of the comparison between PQT and Reinforce presented in the previous lines of experiments by running 5 experiment replicas for each of the 7 tasks using the Neural agent with both training algorithms. We measure an average relative gain of $+1.13\%$ over the final test ROC-AUC achieved by using PQT instead of Reinforce.

We use PQT for the following experiments to train the Evo-NAS and Neural agents to maximize the log likelihood of the top 5 trials. For the Evo-NAS and Evolutionary agents, we have set the mutation probability $p$ to 0.5. Evo-NAS and Neural agents use PQT with learning rate 0.0001 and entropy penalty 0.1. These parameters were selected with a preliminary tuning to ensure a fair comparison.

To measure the quality of the models generated by the three agents, we run 10 architecture search experiment replicas for each of the 7 tasks, and we measure the test ROC-AUC obtained by the best model generated by each experiment replica. The results are summarized in Table 1. The Evo-NAS agent achieves the best performance on all 7 tasks. On 3 out of 7 tasks it significantly outperforms both the Neural and the Evolutionary agents.

Table 1: Best ROC-AUC(%) on the testset for each algorithm and dataset. We report the average over 10 distinct architecture search runs, as well as $\pm$ 2 standard-error-of-the-mean (s.e.m.). Bolding indicates the best performing algorithm or those within 2 s.e.m. of the best.

| Dataset | Neural | Evolutionary | Evo-NAS |
|---|---|---|---|
| 20Newsgroups | $95.45 \pm 0.17$ | $95.31 \pm 0.39$ | $\mathbf{95.67 \pm 0.19}$ |
| Brown | $\mathbf{66.29 \pm 1.44}$ | $\mathbf{66.79 \pm 1.31}$ | $\mathbf{66.82 \pm 1.25}$ |
| ConsumerComplaints | $55.08 \pm 0.97$ | $54.43 \pm 1.41$ | $\mathbf{56.63 \pm 0.71}$ |
| McDonalds | $\mathbf{71.14 \pm 1.19}$ | $\mathbf{71.00 \pm 1.43}$ | $\mathbf{71.90 \pm 1.03}$ |
| NewsAggregator | $\mathbf{99.03 \pm 0.04}$ | $\mathbf{99.01 \pm 0.04}$ | $\mathbf{99.03 \pm 0.04}$ |
| Reuters | $92.36 \pm 0.36$ | $\mathbf{92.68 \pm 0.36}$ | $\mathbf{92.89 \pm 0.21}$ |
| SmsSpamCollection | $99.76 \pm 0.10$ | $99.75 \pm 0.08$ | $\mathbf{99.82 \pm 0.05}$ |

We also report the number of trials each of the agents performed during the 2h long experiments, and summarize the results in Appendix E.5. We find that the Evolutionary and the Evo-NAS agents strongly outperform the Neural agent in terms of number of trials performed. The Evo-NAS agent achieves the largest number of trials on 6 out of 7 datasets, while the Evolutionary agent on 5 out of 7 datasets. On 4 datasets the Evolutionary and Evo-NAS agents perform joint best. This shows that the evolutionary algorithms are biased towards faster models, as shown in (Real et al., 2018).

An in-depth analysis of the architectures that achieved the best performance is beyond the scope of this paper. However, we describe some noticeable patterns, along with more information about the datasets, search space, experiments and setup, in Appendix E.

## 6.4 IMAGE CLASSIFICATION TASK

We also compare the agents on a different architecture search domain: image classification. This is a higher complexity task and the most common benchmark for architecture search (Zoph and Le, 2017; Real et al., 2018; Liu et al., 2017b; 2018).

As shown in recent studies (Zoph et al., 2017; Liu et al., 2017b), the definition of the architecture search space is critical to be able to achieve state-of-the-art performance. In this line of experiments, we reuse the Factorized Hierarchical Search Space defined in (Tan et al., 2018). This is a recently proposed search space that has shown to be able to reach state-of-the-art performance. Note that we use the variant of the search space that does not contain a squeeze-and-excitation block (Hu et al., 2018), so it is more comparable with other works, most of which do not use this component. We abstain from proposing an improved search space that could allow to set a new state-of-the-art, since the main objective of this work is to analyze and compare the properties of the agents.

As the target image classification task we use ImageNet (Russakovsky et al., 2015). As it is common in the architecture search literature, we create a validation set by randomly selecting 50K images from the training set. The accuracy computed on this validation set is used as the reward for the agents, while the original ImageNet test set is used only for the final evaluation.

Following common practice in previous architecture search work (Zoph and Le, 2017; Real et al., 2018; Tan et al., 2018), we conduct architecture search experiments on a smaller proxy task, and then transfer the top-performing discovered model to the target task. As a simple proxy task we use ImageNet itself, but with fewer training steps. During architecture search, we train each generated model for 5 epochs using aggressive schedules for learning rate and weight decay, and then evaluate the model on the 50K validation images.

During a single architecture search experiment, each agent trains thousands of models. However, only the model achieving the best reward is transferred to the full ImageNet. As (Tan et al., 2018), for full ImageNet training, we train for 400 epochs using RMSProp optimizer with decay 0.9 and momentum 0.9, batch norm momentum 0.9997, and weight decay 0.00001. The learning rate linearly increases from 0 to 0.256 in the first 5-epoch warmup training stage, and then decays by 0.97 every 2.4 epochs. We use standard Inception preprocessing and resize input images to $224 \times 224$.

Every architecture search experiment trains 60 generated models in parallel. Each model is trained on a Cloud TPUv2, and takes approximately 3 hours to complete the training on the proxy task. Because of the high cost of experiment, we limit the agents' hyperparameters tuning, and we set them to the values that have worked well in the previous experiments. Evo-NAS and Neural agents use PQT with learning rate 0.0001 and entropy penalty 0.1. PQT maximizes the log likelihood of the top 5% trials. Population size is $P = 500$ and the sample size $S = 50$, for both the Evolutionary and the Evo-NAS agent. The only parameter we do a preliminary tuning for is the mutation probability $p$, since in our experience this parameter is the most sensitive to the complexity of the search space, and the Factorized Hierarchical Search Space used for this experiments is orders of magnitude more complex: it contains $1.6 \cdot 10^{17}$ different architectures. To tune $p$, we run 4 experiments using the Evolutionary agent with values: $p \in \{0.03, 0.05, 0.1, 0.2\}$, and choose the best $p = 0.1$ to be used for both Evo-NAS and Evolutionary agents. Due to the high cost of the experiments, we do not run experiment replicas. Notice that this is common practice for architecture search on image domain.

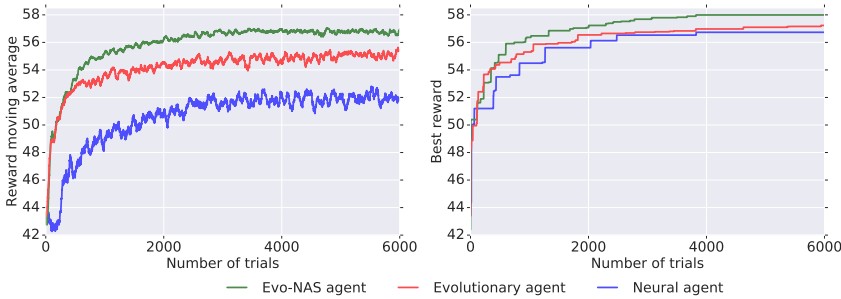

Figure 4: Reward moving average (left) and best reward (right) on the ImageNet proxy task.

In Figure 4 we show the plots of the metrics tracked during the architecture search experiments. Each architecture search experiment required ~304 hours to produce 6000 trials. The plot of the moving average of the reward confirms the properties that we observed in the previous lines of experiments. The Neural agent has a slower start, while Evo-NAS retains the initial sample efficiency of the Evolutionary agent, and is able to improve the quality of the samples generated in the longer

term by leveraging the learning ability. The discussed properties are also visible on the plot of the best reward in Figure 4 (Right). The Neural agent has a slower start, but is able to close the gap with the Evolutionary agent in the longer term. While Evo-NAS shows an initial rate of improvement comparable to the Evolutionary agent, it is able to outperform the other agents in the later stages.

As an additional baseline, we run a Random Search based agent up to 3000 trials. Its reward moving average does not show improvements over time as expected. The max reward achieved is 52.54, while Evo-NAS achieves reward of 57.68 with the same number of trials. These results confirm the complexity of the task. Refer to Appendix F.1 for more details about the comparison with Random Search.

The best rewards achieved by each agent are respectively: 56.73 for the Neural agent, 57.23 for the Evolutionary agent, and 57.99 for Evo-NAS. Trial 2003 of Evo-NAS is the first one that outperforms all models generated by the other agents. Thus, Evo-NAS surpasses the performance of the other agents with only 1/3 of the trials. Furthermore, during the course of the entire experiment, Evo-NAS generates 1063 models achieving higher reward than any model generated by the other agents.

Finally, for each of the agents we select the generated model that achieved the best reward, train them on the full ImageNet task, and evaluate on the held-out test set. This allows to measure the extent to which the reward gains on the proxy task translate to the full task, and also compare with other results published on this benchmark. The results are reported in Table 2, which shows, that the reward gains translate to the final task. Also, the achieved test errors are comparable to the best results published on this benchmark. Notice that this comparison is influenced by factors unrelated to the choice of the agent. For example, some of the methods presented in Table 2 used CIFAR-10 as the proxy task, while others used a subset of ImageNet. The definition of architecture search space is an other important factor in determining the quality of the generated models on the downstream task. MNasNet-92 is the only published result of a network that was generated by exploring the same Factorized Hierarchical Search Space (Tan et al., 2018). It achieves slightly lower results even compared to our Neural agent baseline. Our hypothesis is that this delta can be justified by considering that MNasNet was generated by maximizing a hybrid reward accounting for model latency. Note that although Table 2 presents a variety of different methods, all listed networks have a similar number of parameters of around 5M.

Table 2: Comparison of mobile-sized state-of-the-art image classifiers on ImageNet.

| Architecture | Test error (top-1) | Search cost (gpu days) | Search method |
|---|---|---|---|
| Inception-v1 (Szegedy et al., 2015) | 30.2 | – | manual |
| MobileNet-v1 (Howard et al., 2017) | 29.4 | – | manual |
| ShuffleNet-v2 (Zhang et al., 2017) | 26.3 | – | manual |
| DARTS (Liu et al., 2018) | 26.9 | 4 | gradient |
| NASNet-A (Zoph et al., 2017) | 26.0 | 1800 | rl |
| PNAS (Liu et al., 2017b) | 25.8 | 225 | smbo |
| AmoebaNet-C (Real et al., 2018) | **24.3** | 3150 | evo. |
| MnasNet-92 (Tan et al., 2018) | 25.2 | 988 | rl |
| Neural agent best model | 24.78 | 740 | rl |
| Evolutionary best model | 24.70 | 740 | evo. |
| Evo-NAS best model | **24.57** | 740 | evo. + rl |

The architectures of the best models generated by the 3 agents show noticeable common patterns, which we discuss in Appendix F.2.

# 7 CONCLUSION

We introduce a class of Evo-NAS hybrid agents, which are designed to retain both the sample efficiency of evolutionary approaches, and the ability to learn good architectural patterns of Neural agents. We experiment on synthetic, text and image classification tasks, analyze the properties of the proposed Evo-NAS agent, and show that it outperforms both Neural and Evolutionary agents. Additionally, we show that Priority Queue Training outperforms Reinforce also on architecture search applications. Notice that Evo-NAS is not specific to architecture search. As future work, it would be interesting to apply it to other reinforcement learning tasks.

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

## A  Evo-NAS pseudocode

In Algorithm 1, we formalize our approach in the form of pseudocode.

---

**Algorithm 1:** Sampling algorithm for the Evo-NAS agent

---

Parameters: mutation probability $p \in [0, 1]$, parent sequence $A = (a_1, ..., a_n)$;
Initialize the state of the RNN $h$ to **0**;
Initialize input to the next RNN time-step $x_{in}$ to **0**;
Initialize the output child sequence $A'$ to [ ];
Initialize $logProbability$ to 0;
Initialize $totalEntropy$ to 0;
**for** $i = 1, 2, ..., n$ **do**
    Feed $x_{in}$ to the RNN and run it for one time-step to update $h$ and obtain the output $logits_i$;
    Sample $m \sim \text{Bernoulli}(p)$;
    **if** *m is 0* **then**
        $a'_i = a_i$;
    **else**
        Sample $a'_i$ according to the distribution: $softmax(logits_i)$;
    **end if**
    Append $a'_i$ to $A'$;
    Set $x_{in}$ to the embedding of $a'_i$;
    $logProbability \mathrel{+}= log(p(a'_i|logits_i))$;
    $totalEntropy \mathrel{+}= entropy(softmax(logits_i))$;
**end for**
**return** $A'$ defining the child model;
**return** $logProbability$ to compute the loss for this child model;
**return** $totalEntropy$ to compute the entropy regularization factor to add to the loss;

---

## B  Details on baseline architecture search algorithms

To highlight the properties of the different approaches, we propose to consider two characteristics: 1) whether the agent has learnable parameters, enabling it to learn patterns; 2) whether the agent is capable of efficiently leveraging good past experiences by using mutations. These two characteristics are independent, and for a fixed architecture search algorithm, both of them may or may not be present. In Table 3, we summarize the characteristics of the methods we aim to compare.

Table 3: Properties of the compared architecture search algorithms.

| Algorithm | Learning | Mutation |
|---|---|---|
| Random Search | No | No |
| Neural agent | Yes | No |
| Evolutionary agent | No | Yes |
| Evo-NAS agent | Yes | Yes |

To make comparing the Evo-NAS agent with the baseline agents easier, we provide equivalents of Figure 1. See Figure 5 for the Evolutionary agent, and Figure 6 for the Neural agent.

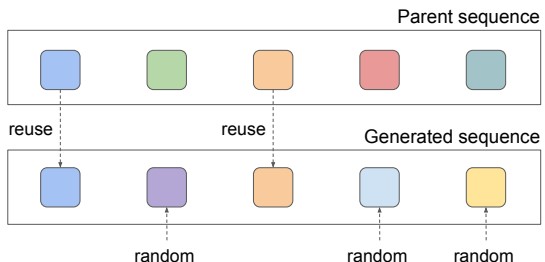

Figure 5: Overview of the how the Evolutionary agent creates a sequence of values for each architectural choice specifying a generated model, given a parent trial. Each of the colored blocks represents an architectural choice set to a specific value. Each action is re-sampled randomly with probability $p$, or reused from the parent with probability $1 - p$.

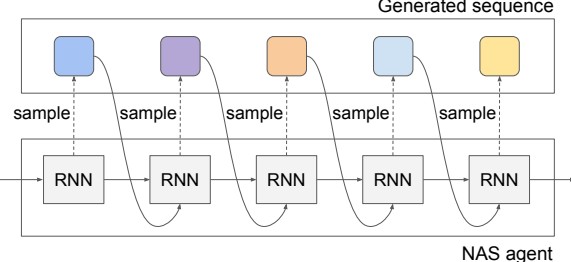

Figure 6: Overview of how the Neural agent samples a trial. Each of the colored blocks represents an architectural choice set to a specific value. Each action is sampled from a distribution defined by an RNN.

## C  DETAILS ON "LEARN TO COUNT" TOY TASK

The proposed toy task has multiple key properties:

- The size of the search space is $n^n$, which even for small $n$ is already too big for any exhaustive search algorithm to succeed.

- As shown by our experiments, Random Search performs very poorly. This allows to attribute the discovery of good sequences to properties of the algorithm, rather than to accidental discovery over time.

  The experimental observation that Random Search performs badly can be intuitively explained as follows. Let $U$ be a uniform distribution over $\{1, \cdots, n\}^n$, then:

$$\mathop{\mathbb{E}}_{a \sim U} \left[ r(a)^{-1} \right] \in O\left(n^2\right)$$

  This suggests that, for a random sequence $\mathbf{a}$, $r(\mathbf{a})$ can be expected to be much smaller than 1, which is again confirmed by the experiments.

- The search space exhibits a sequential structure, with a notion of locally optimal decisions, making it a good task both for learning patterns, and for mutating past trials by local modifications.

Before conducting the experiments on the toy task, we tuned hyperparameters of all the agents to ensure a fair comparison. We set the mutation probability $p = 0.3$, the population size $P = 500$, and the sample size $S = 50$, for both the Evolutionary and the Evo-NAS agent. We set the learning rate to be 0.0005 for the Neural agent and 0.0001 for the Evo-NAS agent. We set the entropy penalty to 0.1 for the Neural agent and 0.2 for the Evo-NAS agent. PQT maximizes the log likelihood of the top 5% trials for the Neural agent and top 20% trials for the Evo-NAS agent.

## D  DETAILS ON NASBENCH

We used the hyperparameters from the synthetic task experiment as a starting point, and optimized for mutation probability, learning rate and entropy penalty if applicable. In our experiment, PQT maximizes the log likelihood of top 20% trials for both Neural and Evo-NAS agents.

The hyperparameters were optimized using the following procedure:

1 Start with the hyperparameters used in Section 6.1 and scan learning rate: $[0.0001, 0.0002, 0.0005, 0.001]$ and entropy penalty: $[0.1, 0.2, 0.5, 1.0]$ for the best moving average reward with window size 50.

2 Use the best learning rate and entropy penalty from step (1) and scan mutation probability: $[0.4, 0.3, 0.2, 0.1]$ for the best moving average reward with window size 50.

3 Repeat step (1) but scan learning rate: $[0.00001, 0.00005, 0.0001, 0.0002]$ and entropy penalty: $[0.01, 0.05, 0.1, 0.2]$.

The resulting hyperparameters are collected in Table 4.

Table 4: Optimized hyperparameters for Random, Evolutionary, Neural (PQT/Reinforce) and Evo-NAS (PQT/Reinforce) agents in NASBench.

| Agent | Mutation probability | Learning rate | Entropy penalty |
|---|---|---|---|
| Evo-NAS (PQT) | 0.2 | 0.00005 | 0.01 |
| Evo-NAS (Reinforce) | 0.2 | 0.0002 | 0.01 |
| Neural (PQT) | - | 0.001 | 0.1 |
| Neural (Reinforce) | - | 0.001 | 0.1 |
| Evolutionary | 0.3 | - | - |
| Random search | - | - | - |

# E    DETAILS ON TEXT CLASSIFICATION EXPERIMENTS

## E.1    DATASETS

Table 5: References for the datasets used in the text experiments.

| Dataset | Reference |
|---|---|
| 20Newsgroups | (Lang, 1995) |
| Brown | (Francis and Kuera, 1982) |
| ConsumerComplaints | catalog.data.gov |
| McDonalds | crowdflower.com |
| NewsAggregator | (Lichman, 2013) |
| Reuters | (Debole and Sebastiani, 2004) |
| SmsSpamCollection | (Almeida et al., 2011) |

Table 6: Statistics of the text classification tasks.

| Dataset | Train samples | Valid samples | Test samples | Classes | Lang | Mean text length |
|---|---|---|---|---|---|---|
| 20 Newsgroups | 15,076 | 1,885 | 1,885 | 20 | En | 2,000 |
| Brown Corpus | 400 | 50 | 50 | 15 | En | 20,000 |
| Consumer Complaints | 146,667 | 18,333 | 18,334 | 157 | En | 1,000 |
| McDonalds | 1,176 | 147 | 148 | 9 | En | 516 |
| News Aggregator | 338,349 | 42,294 | 42,294 | 4 | En | 57 |
| Reuters | 8,630 | 1,079 | 1,079 | 90 | En | 820 |
| SMS Spam Collection | 4,459 | 557 | 557 | 2 | En | 81 |

## E.2    SEARCH SPACE

For our text classification experiments, we designed a search space of two-tower models, similar to the one used by (Wong et al., 2018). One tower is a deep FFNN, built by stacking: a pre-trained text-embedding module, a stack of fully connected layers, and a softmax classification layer. The other tower is a wide-shallow layer that directly connects the one-hot token encodings to the softmax classification layer with a linear projection. The wide tower allows the model to learn task-specific biases for each token directly, such as trigger words, while the deep tower allows it to learn complex patterns. The wide tower is regularized with L1 loss to promote sparsity, and the deep tower is regularized with L2 loss. We provide additional details in Table 7.

The agent defines the generated model architecture by selecting a value for every available architectural or hyperparameter choice. The first action selects a pre-trained text-embedding module. The details of the text-embedding modules are reported in Table 8. These modules are available via the TensorFlow Hub service[1]. Using pre-trained text-embedding modules has two benefits: first, improves the quality of the generated models trained on smaller datasets, and second, decreases convergence time of the generated models.

The optimizer for the deep column can be either Adagrad (Duchi et al., 2011) or Lazy Adam [2]. "Lazy Adam" refers to a commonly used version of Adam (Kingma and Ba, 2014) that computes the moving averages only on the current batch. These are efficient optimizers, that halve the back-propagation time, compared to more expensive optimizers such as Adam. The optimizer used for the wide column is FTRL (McMahan, 2011).

---

[1] https://www.tensorflow.org/hub
[2] https://www.tensorflow.org/api_docs/python/tf/contrib/opt/LazyAdamOptimizer

Table 7: The search space defined for text classification experiments.

| Parameters | Search space |
|---|---|
| 1) Input embedding modules | Refer to Table 8 |
| 2) Fine-tune input embedding module | {True, False} |
| 3) Use convolution | {True, False} |
| 4) Convolution activation | {relu, relu6, leaky relu, swish, sigmoid, tanh} |
| 5) Convolution batch norm | {True, False} |
| 6) Convolution max ngram length | {2, 3} |
| 7) Convolution dropout rate | {0.0, 0.1, 0.2, 0.3, 0.4} |
| 8) Convolution number of filters | {32, 64, 128} |
| 9) Number of hidden layers | {0, 1, 2, 3, 5} |
| 10) Hidden layers size | {64, 128, 256} |
| 11) Hidden layers activation | {relu, relu6, leaky relu, swish, sigmoid, tanh} |
| 12) Hidden layers normalization | {none, batch norm, layer norm} |
| 13) Hidden layers dropout rate | {0.0, 0.05, 0.1, 0.2, 0.3, 0.4, 0.5} |
| 14) Deep optimizer name | {adagrad, lazy adam} |
| 15) Lazy adam batch size | {128, 256} |
| 16) Deep tower learning rate | {0.001, 0.005, 0.01, 0.05, 0.1, 0.5} |
| 17) Deep tower regularization weight | {0.0, 0.0001, 0.001, 0.01} |
| 18) Wide tower learning rate | {0.001, 0.005, 0.01, 0.05, 0.1, 0.5} |
| 19) Wide tower regularization weight | {0.0, 0.0001, 0.001, 0.01} |
| 20) Number of training samples | {1e5, 2e5, 5e5, 1e6, 2e6, 5e6} |

Table 8: Options for text input embedding modules. These are pre-trained text embedding tables, trained on datasets with different languages and size. The text input to these modules is tokenized according to the module dictionary and normalized by lower-casing and stripping rare characters. We provide the handles for the modules that are publicly distributed via the TensorFlow Hub service (https://www.tensorflow.org/hub).

| Language/ID | Dataset size (tokens) | Embed dim. | Vocab. size | Training algorithm | TensorFlow Hub Handles Prefix: https://tfhub.dev/google/ |
|---|---|---|---|---|---|
| English-small | 7B | 50 | 982k | Lang. model | nnlm-en-dim50-with-normalization/1 |
| English-big | 200B | 128 | 999k | Lang. model | nnlm-en-dim128-with-normalization/1 |
| English-wiki-small | 4B | 250 | 1M | Skipgram | Wiki-words-250-with-normalization/1 |
| Universal-sentence-encoder | - | 512 | - | (Cer et al., 2018) | universal-sentence-encoder/2 |

### E.3 ARCHITECTURAL PATTERNS

Here, we mention a few relevant patterns of the best architectures generated across tasks. The FFNNs for the deep part of the network are often shallow and wide. The learning rate for both wide and deep parts is in the bottom of the range (0.001). The L1 and L2 regularization are often disabled. Our interpretation of this observation is that reducing the number of parameters is a simpler and more effective regularization, which is preferred over adding L1 and L2 factors to the loss.

### E.4 CHOICE OF METRIC

For the text classification experiments, we used ROC-AUC instead of the more commonly used accuracy, since it provides a less noisy reward signal. In a preliminary experiment, we validated this hypothesis by running experiments on the ConsumerComplaints task. Then, for a sample of 30 models, we have computed 4 metrics: ROC-AUC on validation and test set, accuracy on validation and test set. The Pearson correlation between the validation ROC-AUC and the test ROC-AUC was 99.96%, while between the validation accuracy and the test accuracy was 99.70%. The scatter plot of these two sets is shown in Figure 7.

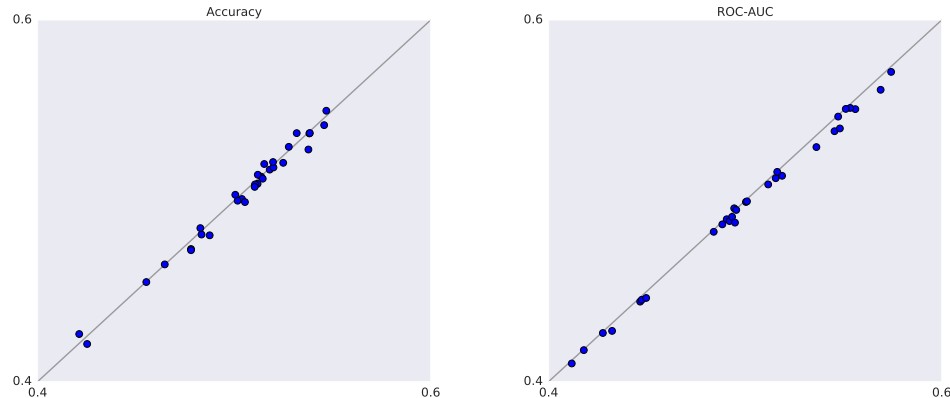

Figure 7: Correlation of validation accuracy with test accuracy (Left) and validation ROC-AUC with test ROC-AUC (Right). The correlation is higher for ROC-AUC. For plotting the correlations, we used the ConsumerComplaints dataset.

### E.5 COMPARISON OF NUMBER OF TRIALS PERFORMED

Table 9: The number of trials performed for the experiments from Figure 9. We report the average over 10 runs, as well as $\pm$ 2 standard-error-of-the-mean (s.e.m.). Bolding indicates the algorithm with the highest number of trials or those that have performed within 2 s.e.m. of the largest number of trials.

| Dataset | Neural | Evolutionary | Evo-NAS |
|---|---|---|---|
| 20Newsgroups | $136 \pm 6$ | $\mathbf{175 \pm 11}$ | $\mathbf{174 \pm 10}$ |
| Brown | $109 \pm 4$ | $\mathbf{119 \pm 6}$ | $\mathbf{116 \pm 8}$ |
| ConsumerComplaints | $122 \pm 5$ | $\mathbf{150 \pm 9}$ | $142 \pm 12$ |
| McDonalds | $213 \pm 21$ | $258 \pm 27$ | $\mathbf{359 \pm 65}$ |
| NewsAggregator | $304 \pm 33$ | $\mathbf{342 \pm 32}$ | $291 \pm 26$ |
| Reuters | $160 \pm 8$ | $\mathbf{194 \pm 13}$ | $\mathbf{201 \pm 19}$ |
| SmsSpamCollection | $390 \pm 58$ | $410 \pm 41$ | $\mathbf{617 \pm 150}$ |

### E.6 ADDITIONAL DETAILS

In Figure 9, we visually present the results of the experiments that we aggregated in Table 1.

### E.7 TRENDS OVER TIME

To verify that the learning patterns highlighted in Section 6.1 generalize, we plot in Figure 10 the reward moving average for two tasks: 20Newsgroups and ConsumerComplaints. For these experiments, we have extended the time budget from 2h to 5h. This time budget extension is needed to be able to capture long-term trends exhibited by the Neural and Evo-NAS agents. We run 3

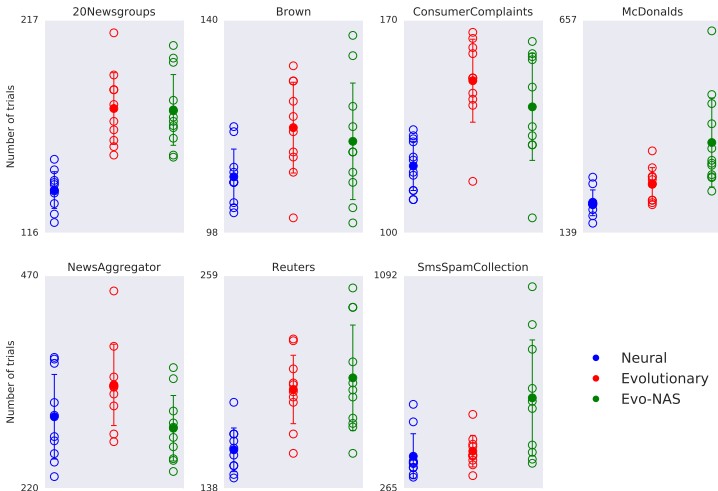

Figure 8: Number of trials performed for the experiments from Figure 9. The empty circles represent the number of trials performed in each of the 10 experiment replicas. The filled circles represent the means of the empty circles. We superpose ±1 standard deviation bars.

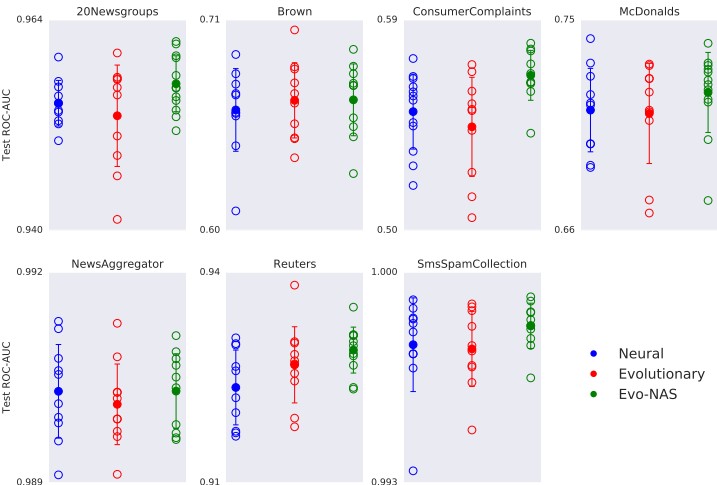

Figure 9: Results of the experiments on 7 text classification tasks. Each experiment was run 10 times. For each run, we have selected the model that obtained the best ROC-AUC on the validation set (the best reward). These best models were then evaluated by computing the ROC-AUC on the holdout testset. The empty circles in the plot represent the test ROC-AUC achieved by each of the 10 best models. The filled circles represent the means of the empty circles. We superpose ±1 standard deviation bars.

replicas for each task. In the early stages of the experiments, we notice that the quality of the samples generated by the Neural agent are on the same level as the randomly generated samples, while the quality of the samples generated by the Evo-NAS and Evolutionary agents grows steadily. In the second half of the experiments, the Neural agent starts applying the learned patterns to the generated samples. The quality of the samples generated by the Evolutionary agent flattens, which we assume is due to the fact that the quality of the samples in the population is close to optimum, and the quality of the samples cannot improve, since good mutations patterns cannot be learned. Finally, we observe that the Evo-NAS agent keeps generating better samples.

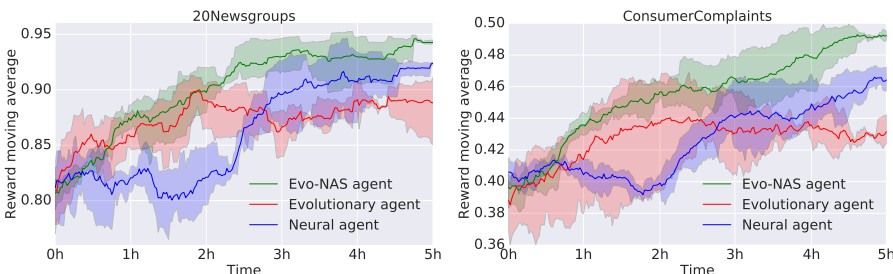

Figure 10: Reward moving average for the compared agents. The average is computed over a window of 50 consecutive trials. We ran 3 replicas for each experiment. The shaded area represents minimum and maximum value of the rolling average across the runs.

## F    DETAILS ON IMAGE EXPERIMENTS

### F.1    COMPARSION WITH RANDOM SEARCH

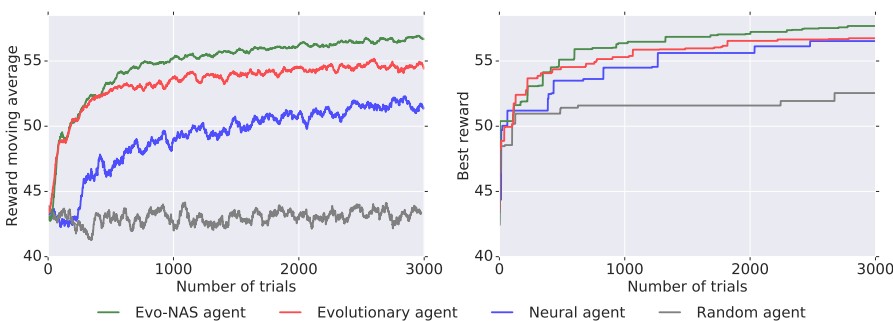

Figure 11:  Quality metrics of the different agents during the first 3k trials of architecture search on the image classification proxy task. The reward is computed on the validation set, while the test set is used only for the final evaluation of the selected network. We report the moving average of the reward over 50 trials (Left) and the best reward attained so far (Right).

### F.2    ARCHITECTURAL PATTERNS

The core of all 3 networks is mostly constructed with convolutions with kernel size 5 by 5, and have similar network depth of 22 or 23 blocks. The networks found b3y Evolutionary and Evo-NAS agents both have same first and last block, but Evo-NAS tends to use more filters (such as 192 and 384) in later stages to achieve higher accuracy than the Evolutionary agent. We show the best networks in Figure 12.

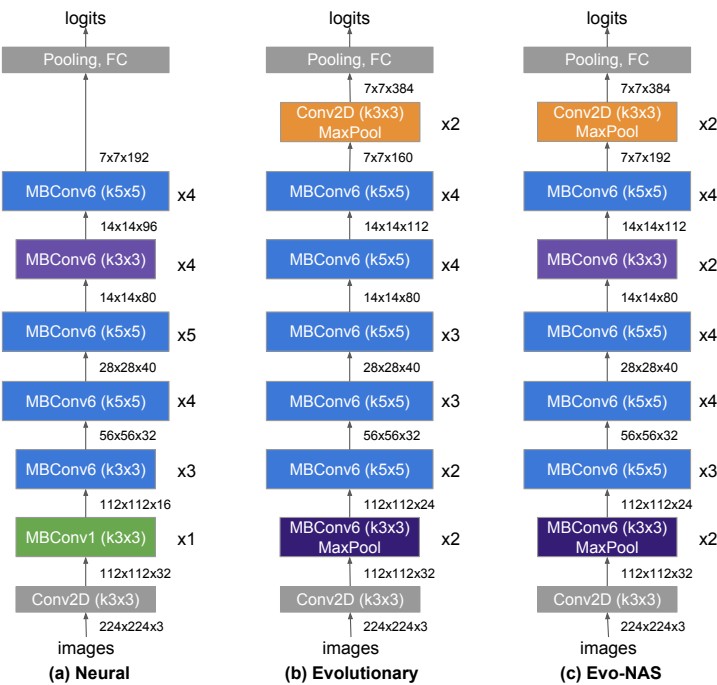

Figure 12: Neural networks achieving the best reward for image classification generated by: (a) Neural agent, (b) Evolutionary agent, (c) Evo-NAS agent. For a detailed description of the Factorized Hierarchical Search Space and its modules refer to (Tan et al., 2018).

