# OpenReview forum: "Evo-NAS: Evolutionary-Neural Hybrid Agent for Architecture Search"
_ICLR.cc/2020/Conference — Reject_

### Official Review · AnonReviewer1 · 2019-10-23
**Official Blind Review #1**

**Rating:** 6

**Review:**

It is a nice paper that combines the deep reinforcement learning and evolutionary learning techniques to neural architecture search problem. Experimental results are promising. However, I still have some concerns on the current submission.
1.In Fig 1,2 &3, it seems that the performances of Neural (PQT) keeps increasing. For better compassion, we recommend the authors reports the performances of compared algorithms until they are convergent.
2.The different training algorithms (Reinforce and PQT) have difference performances whether because different training algorithms converge to difference local minima or stationary points.


**Experience Assessment:**

I have read many papers in this area.

**Review Assessment: Checking Correctness Of Derivations And Theory:**

I assessed the sensibility of the derivations and theory.

**Review Assessment: Checking Correctness Of Experiments:**

I assessed the sensibility of the experiments.

**Review Assessment: Thoroughness In Paper Reading:**

I made a quick assessment of this paper.

---

> ### Author Response · Authors · 2019-11-07
> **Thank you for your review**
>
> Thank you for you comments! Below we specifically address both of the points that you raised. Let us know if you have any more concerns - we are happy to make improvements to the paper.
>
> 1. To make the assessment that the methods have converged, we looked at the best reward attained (right plots), not reward running average (left plots). By “reward running average”, we mean sliding a constant size window over the reward, and computing the average reward in each such window. This measure is not a good metric to compare the algorithms, and we use it just to show how much the reward varies from sample to sample for different methods.
>
> Note that learning-based methods (like Neural PQT which you mentioned) do see small improvements in running average reward even long after they stop exploring new parts of the search space - this is simply because they keep making their distributions over actions “sharper”. Therefore, it is more informative to look at best reward attained over time.
>
> With that in mind, we review below the experiments that you pointed to - that is, Figures 2 and 3. You also mentioned Figure 1, which does not correspond to any experiment, so we assumed this to be a typo - but let us know if you meant something else there, and we can discuss further.
>
> - For NAS-Bench (Figure 3), the methods see only marginal improvements at the end of the experiment.
>
> - For the synthetic task (Figure 2), all methods see small improvements of the reward if ran longer, but the overall pattern remains the same (Evo-NAS (PQT) attaining the best results, matched by Neural (PQT) from around 4500 trials onward).
>
> 2. Our understanding is that PQT provides a stronger training signal than Reinforce. This comes at a cost, since Reinforce is known to be unbiased, and therefore it produces gradients that (in expectation) match the gradient of the expected reward. On the other hand, PQT is a heuristic, and it gives no theoretical guarantees.
>
> We wouldn’t say that PQT is always better than Reinforce - this is likely very domain dependent. Our focus is architecture search, where the number of trials is typically low given the complexity of the search spaces. In this domain, as seen in our experiments, trading off theoretical guarantees and lack of bias for a stronger and more greedy training signal leads to better results.
>
> Nevertheless, comparing PQT to classical RL algorithms (like Reinforce, but also many others) is a very interesting research direction. Since our main goal was hybridizing RL-based and Evolution-based approaches to architecture search, we did not explore this direction in our work.

---

### Official Review · AnonReviewer3 · 2019-10-24
**Official Blind Review #3**

**Rating:** 3

**Review:**

This paper is well organized. The applied methods are introduced in detail. But it lacks some more detailed analysis.

My concerns are as follows.
1. The performance differences between Evolutionary agent and EVO-NAS agent seem not significant. Please conduct additional statistical tests such as the Wilcoxon signed-rank test to verify the performance improvements are significant.
2. Many studies have been conducted to automatically adjust control parameters such as crossover and mutation probabilities in evolutionary algorithm literature. It would be better to compare one of these approaches in the experiments.

**Experience Assessment:**

I have read many papers in this area.

**Review Assessment: Checking Correctness Of Derivations And Theory:**

I assessed the sensibility of the derivations and theory.

**Review Assessment: Checking Correctness Of Experiments:**

I assessed the sensibility of the experiments.

**Review Assessment: Thoroughness In Paper Reading:**

I read the paper at least twice and used my best judgement in assessing the paper.

---

> ### Author Response · Authors · 2019-11-07
> **Thank you for your review**
>
> Thank you for your review! We address each of your concerns below. Please let us know if you have some further comments - we are happy to make changes to the paper to improve it.
>
> 1. While the gains are not enormous, we agree with Reviewer #2 in saying that they are consistent across a range of different tasks: both in real-life AS scenarios (NAS-Bench, NLP, ImageNet), and in the synthetic task. However, making sure that differences between results are statistically significant is certainly an important point.
>
> For example, note that for NLP (Table 1), we ran 10 runs for each (approach, dataset) pair. We claim that Evo-NAS matches the better of the baselines in all cases, and in 3 out of 7 it outperforms the best baseline with statistical significance. We made this claim for the three datasets where the difference exceeded twice the standard-error-of-the-mean. This can also be easily translated into p-values: for example, for the ConsumerComplaints dataset, the p-value for Evo-NAS’s gains over either baseline being statistically significant is < 0.02 (so we reject the null hypothesis that there is no gain).
>
> We used simple statistical measures such as the ones described above to assess significance. However, we are open to performing additional nonparametric tests, such as the Wilcoxon Signed-Rank Test you proposed. Please let us know what would be the exact test that you had in mind (i.e. what would be the null hypothesis). For example, we could consider pairs of (average result of Evolution, average result of Evo-NAS) over all benchmarks considered in the paper (there are 10 in total since we should count each NLP dataset separately), and perform the test on these pairs. In that case, the Signed-Rank test does show significance, as in all cases the mean result of Evo-NAS is strictly better than the mean result of Evolution.
>
> 2. You are right in saying that there are many smart evolutionary algorithms out there. However, to our knowledge, the NAS domain is dominated by “non-adaptive” evolution (i.e. with uniform prior over mutations), which also seems to perform very well across different AS domains. Since here we focus on architecture search, we compare our method to two strong, common and most recent baselines (the combination of which gives Evo-NAS).
>
> Note that our hybrid may not necessarily be the best algorithm of this kind; our intention was to show that combining these two worlds (learning-based and evolution-based) in a relatively simple way we can get consistent gains across multiple domains.
>
> One kind of follow-up works that we wish to inspire is exactly what you are proposing: bringing different “smart” genetic algorithms from the evolution community to the NAS community, and investigating their behavior on architecture search. It is surely an exciting direction, although one that is not the focus of our current work.

---

### Official Review · AnonReviewer2 · 2019-10-27
**Official Blind Review #2**

**Rating:** 3

**Review:**

In this paper, the authors proposed to combine both NN-based NAS and Aging EVO to get the benefit of both world: good global and local sample efficiency. The main idea is to use Aging EVO algorithm to guide the overall search process, and a NN predicting the final performance is used to guide the mutation process. The combined EVO-NAS has showed consistent good performance over a range of tasks.

Overall, while the novelty of the paper is not exceptional, since it is a rather straightforward combination of two existing approaches, the end-results is promising. I would like to see more in-depth analysis on the combined algorithm to validate authors' hypothesis on why EVO-NAS works better. More detailed comments can be found below.

1. The experiment on the synthetic task is not very helpful, since the domain can be far apart from the real NAS applications. One evidence is that the NN significantly outperform EVO in this task but not in the other tasks.

2. In difficult tasks, the proposed EVO-NAS and the original Aging EVO are very close in the first few hundreds of trials, however, later the gap remains the same. It would be interesting to see if we can eliminate the gap by adding the NN component in the middle of the Aging EVO experiment (e.g., at the point of 2000 trials).

3. I am also curious about the difference between the NN learned from Neural agent vs. those learned from EVO-NAS agent. Moreover, do we need a NN for the EVO-NAS? Or something simpler would be sufficient to guide the search.

**Experience Assessment:**

I have read many papers in this area.

**Review Assessment: Checking Correctness Of Derivations And Theory:**

N/A

**Review Assessment: Checking Correctness Of Experiments:**

I assessed the sensibility of the experiments.

**Review Assessment: Thoroughness In Paper Reading:**

I read the paper at least twice and used my best judgement in assessing the paper.

---

> ### Author Response · Authors · 2019-11-07
> **Thank you for your review**
>
> Thank you for your comments and suggestions! Please let us know if you have any more feedback about our work, since we would be happy to use it to make the paper better. Below we specifically address each of the points that you made.
>
> 1. We agree that the synthetic task is far from real NAS applications. In our paper we do focus on NAS, and the other lines of experiments (NAS-Bench, NLP datasets, ImageNet) are all real-life architecture search scenarios. However, as we hint in the “Conclusion” section, it is possible to apply Evo-NAS to tasks outside of architecture search. Discrete optimization problems such as our toy task could be one example, and we see that Evo-NAS also performs very well in this domain. We think that makes the approach even more promising in terms of inspiring future work.
>
> 2. As we understand the proposal, you would consider running the NN learning component not from the beginning of the experiment, but start learning later on. This would save some computational cost; are we missing any other advantages of this approach? Our initial intuition is that the savings might be limited due to low overhead of running the NN compared to the training of candidate architectures. However, it would be interesting to experiment further in this direction in the future.
>
> 3. You made a good point that we could use something simpler than a Neural Network for the learning part of Evo-NAS. Since we wanted our approach to be comparable to existing methods, we used a NN, since it’s somewhat standard in the NAS community. Also, from a theoretical standpoint, a NN has a useful property of being a universal approximator. On the other hand, it does seem likely that guiding the evolutionary search (i.e. producing a good prior over mutations) might be an easier problem than learning the structure of all solutions (i.e. producing a good prior over the entire search space).
>
> Therefore, it’s certainly possible that using something simpler as the learning part of Evo-NAS would be sufficient to get good results. However, again note that in terms of computational cost the NN part is negligible - so any gains would not be in terms of compute, but perhaps in enhanced explainability of the learning component.

---

> > ### Comment · AnonReviewer2 · 2019-11-07
> > **additional comparison should be provided**
> >
> > Thanks for the comment.
> >
> > After reading others' comment, it seems that similar ideas have been published, e.g., RENAS. I think it is important to compare against these newer baselines with similar ideas to fully justify the contribution, which in my opinion is the consistent good empirical result.

---

> > > ### Author Response · Authors · 2019-11-08
> > > **Comparison with RENAS**
> > >
> > > We agree that the parallel work of RENAS is definitely related to Evo-NAS. Below we review some key differences and reasons why we think the main contributions of our work still stand. While both works hybridize RL with evolution, we believe the goals are very different, and the insights complementary. Please let us know your thoughts on this comparison.
> > >
> > > 1) We believe that the purpose of RENAS was achieving SotA accuracy on CIFAR10 and ImageNet with low computational cost. To that end, they proposed a framework which does use both RL and Evolution to produce child models. However, RENAS is also augmented with several domain-specific optimizations and improvements that could be complementary to many possible search algorithms, including:
> > > - a complex way of proposing new mutations by using attention and a bidirectional LSMT encoder ran on the architecture description
> > > - reloading weights from the parent model into the child model (related to the approach in ENAS)
> > > - a refined reward function: instead of using validation accuracy directly, it is rescaled and passed through the tan() function
> > > - improved search space (not reused from any prior work)
> > >
> > > We think that likely all of these different ideas, together with combining RL with Evolution, contribute to RENAS achieving good results. In contrast, our aim was not to necessarily push Evo-NAS to its limits in terms of accuracy or computational savings - in fact, the improvements listed above could also be applied to Evo-NAS. We explicitly chose not to do that (for example, not to reload child weights or tune the search space), because our goal was to study how RL and evolution work together. The RL and Evolutionary baselines that we compare with are simply ablations of Evo-NAS, so the fact that Evo-NAS outperforms them both really shows that there is something to be gained from both worlds.
> > >
> > > 2) Since our aim was to show that combining learning and evolution leads to gains across domains, we experiment with many: image (ImageNet, NAS-Bench), NLP, and synthetic. While we focus on architecture search, we also believe that our results hint that Evo-NAS would show strong performance in different (although similar) discrete optimization problems - and we would be happy to see follow-up works of this kind. In contrast, RENAS approach strongly focuses on the image domain, and applying it to other domains is a future research opportunity.
> > >
> > > 3) Evo-NAS learns a policy directly on the architecture search-space, while RENAS learns it over the space of possible mutations applicable to a given parent model. Thus, the application of RENAS requires to define an additional mutation grammar and a more complex agent (e.g. composed of 4 distinct mutation generation sub-steps).

---

> > > > ### Comment · AnonReviewer2 · 2019-11-15
> > > > **Thanks for the follow ups**
> > > >
> > > > I would like to thank the authors for the detailed followup comment. Especially the clarification on the difference with RENAS. The goal on studying how RL and evolution work together is great and should be encouraged. However, in my opinion, the totality of the contribution is limited by the scope of the paper.
> > > >
> > > > To improve upon the current version, one possible way is to use the understanding to build better NAS frameworks (on top of the domain specific adaption as mentioned in the comment).

---

> > > > > ### Author Response · Authors · 2019-11-15
> > > > > **Thanks for the response**
> > > > >
> > > > > Thank you for your comment! Could you elaborate on what you mean by "limited by the scope of the paper" and "better NAS frameworks"? Our understanding is that Evo-NAS is fairly general, so in principle it can be applied to any architecture search space.

---

> > > > > > ### Comment · AnonReviewer2 · 2019-11-15
> > > > > > **thanks**
> > > > > >
> > > > > > My overall concern is that the combination of Evolutionary NAS algorithm and RL (or other model to guide the mutation) existed before this work. The contribution of a more detailed study on the combination and/or a slight variant of the combination is not significant enough, in my opinion. Let  me know if I misunderstood anything, thanks.

---

> > > > > > > ### Author Response · Authors · 2019-11-15
> > > > > > > **Evo-NAS contribution**
> > > > > > >
> > > > > > > Thanks for the swift response. RENAS is the only Evolutionary-RL architecture search hybrid that we are aware of, and the approach it took does lead to good performance. However, we believe that the differences between RENAS and Evo-NAS are fundamental.
> > > > > > >
> > > > > > > As we wrote in point 3 of our "Comparison with RENAS" comment, RENAS uses an additional mutation grammar (so, a mutation search space), and the task for RL is to navigate that space. Because of that, we don't see a straightforward way to create an ablation of RENAS containing just the RL part alone (and such an ablation is not covered in the RENAS paper).
> > > > > > >
> > > > > > > On the other hand, Evo-NAS lies on a continuous spectrum. Two extreme points of that spectrum are Evo-NAS's ablations: either having just the evolutionary part, or just the RL part. We believe that, as far as advocating for further exploring this spectrum goes, our paper makes a much stronger point for it than the RENAS paper.

---

### Public Comment · ~Linnan_Wang1 · 2019-09-26
**Fig 3 is misleading**

Hello there,

 In your figure Fig.3(right), random search seems stuck in a local optimum, reflected by the plateau. I suspect this figure is not from sufficient runs. In theory, random search should keep consistently increasing until converging with Evo-NAS; and it will find the best in expected N/2 trails (N is the size of NASBench). It is questionable for random search to reach plateau in only 5000 trails on a dataset that has 420K samples. Thank you.

---

> ### Author Response · Authors · 2019-10-03
> **Good catch**
>
> Thank you for your comment!
>
> Figure 3 (right) actually shows the best moving average of reward over time (i.e. first the moving average was applied, and then maximum over time was computed). This lowered the results for all agents (especially for Random Search) and made Figure 3 not directly comparable to the results from the NasBench paper. Note that RS is not stuck, it's just progressing very slowly.
>
> All the other "best reward" figures in our paper correctly show the best reward over time.
>
> Thank you for pointing out this discrepancy. As soon as it is possible to update the paper, we will upload a version with Figure 3 regenerated. Apart from showing best reward, we will also convert it into the log regret format used in the original NasBench paper.

---

> > ### Public Comment · ~Linnan_Wang1 · 2019-11-06
> > **Thank you**
> >
> > Thank you for your response. It will be great if the authors can compare their algorithms to prior works (LaNAS [1], AlphaX[2] and several peer submissions [3][4]) on NASBench following a similar standard, i.e. using the metric of #samples to global optimum to fairly evaluate the search efficiency.
> >
> > [1] Wang, Linnan, et al. "Alphax: exploring neural architectures with deep neural networks and monte carlo tree search." arXiv preprint arXiv:1805.07440 (2018).
> >
> > [2] Wang, Linnan, et al. "Sample-Efficient Neural Architecture Search by Learning Action Space." arXiv preprint arXiv:1906.06832 (2019).
> >
> > [3] https://openreview.net/forum?id=rJgffkSFPS
> >
> > [4] https://openreview.net/forum?id=B1lxV6NFPH

---

### Public Comment · ~Qingquan_Song1 · 2019-11-06
**Potential Baseline to Compare with?**

Hello there,

          Nice paper! It would be great to provide some comparison or introduction of a related work:

RENAS: Reinforced Evolutionary Neural Architecture Search (CVPR 2019)

since it also proposed a mixture approach to conjoint the advantage of reinforcement learning and evolutionary method.

---

> ### Author Response · Authors · 2019-11-06
> **Thank you for pointing out a related paper**
>
> Thanks for directing us to this work, it does look related to our approach. After reviewing the paper, it looks like it also hybridizes learning and evolution, although in a somewhat complicated way, making it harder to reason about the source of accuracy gains. Nevertheless, we are glad to see that these kinds of ideas are an active area of research. When we update our paper, we will mention RENAS in the "Related Works" section.

---

### Author Response · Authors · 2019-11-14
**Paper revision after all the reviews and comments**

We would like to thank all reviewers and commenters again for their valuable feedback!

We have just uploaded an updated version of our paper, which includes two changes:
- We updated the NAS-Bench figure in response to comments from Linnan Wang. We also switched to reporting regret instead of accuracy, and using log scales on both axis - this matches “NAS-Bench-101: Towards Reproducible Neural Architecture Search”), making Figure 7 in that paper directly comparable with our new Figure 3.
- We expanded the “Related Works” section to describe differences between our work and RENAS.

For a more detailed comparison of Evo-NAS with RENAS, see our comment “Comparison with RENAS” below. Note that we have recently updated that comment (after reviewing RENAS more closely) to include another important difference (number 3).

---

### Decision · Program_Chairs · 2019-12-19

**Decision:**

Reject

**Comment:**

Thanks to the authors for the revision and discussion. This paper provides a neural architecture search (NAS) method, called Evolutionary-Neural hybrid agents (Evo-NAS), which combines NN-based NAS and Aging EVO. While the authors' response addressed some of the reviewers' comments, during discussion period there is a new concern that the idea proposed here highly overlaps with the method of RENAS, which stands for Reinforced Evolutionary Neural Architecture Search. Reviewers acknowledge that this might discount the novelty of the paper. Overall, there is not sufficient support for acceptance.